# The Use and Complications of Halo Gravity Traction in Children with Scoliosis

**DOI:** 10.3390/children9111701

**Published:** 2022-11-06

**Authors:** Mihai B. Popescu, Alexandru Ulici, Madalina Carp, Oana Haram, Nicolae S. Ionescu

**Affiliations:** 1Emergency Hospital for Children “Grigore Alexandrescu”, 011743 Bucharest, Romania; 211th Department, “Carol Davila” University of Medicine and Pharmacy, 050474 Bucharest, Romania; 3Emergency Hospital for Children “Maria Sklodowska Curie”, 011743 Bucharest, Romania

**Keywords:** scoliosis, halo, traction, complications, children

## Abstract

Scoliosis is one of the most frequent spine deformities encountered in children and is regularly discovered after 15 years of age with a girls to boys ratio of 2:1. Vertebral arthrodesis involves both short and long term complications. Neurological complications consist of nerve root injuries, cauda equina or spinal cord deficit. Traction is a good orthopaedic technique of progressive deformity correction which attempts to minimize complications. The purpose of this study is to assess the complications that arise during halo gravity traction and to evaluate the correction of the scoliotic curves under traction. A single centre prospective study was conducted on 19 paediatric patients suffering from scoliosis that were admitted between 2019–2022. Traction-related complications were encountered in 94.7% of patients, with the most frequent being cervical pain (89.5%). It was followed by back pain, in 36.8% of the cases, with just 5.3% of the cases having experienced vertigo or pin displacement. Neurological symptoms were present in 26.3% of the patients and pin pain and pin infection equally affected 26.3% of patients. Even though minor halo related complications are frequent, with proper patient monitoring they can be addressed, thus making traction a safe method for progressive curve correction.

## 1. Introduction

Scoliosis is one of the most frequent spine deformities encountered in children, ranging between 0.47% (Turkey) to 5.2% (Germany) for idiopathic scoliosis [1,2,3]. It is regularly encountered in children older than 15 with a girls to boys ratio of 2:1 [1]. The etiology of scoliosis is not well known and the risk factors associated with idiopathic scoliosis are unclear. A correlation between developmental dysmorphism and scoliosis has been proven. Even though physical activities are encouraged, rhythmic gymnastics, ballet or dance can be associated with idiopathic scoliosis [4]. The risk for curve progression is higher in girls, especially before the menses and for the ones with right thoracic and double curves, while boys with right lumbar curves have a greater risk of advancement. In addition, curve progression was found to be larger before the pubertal growth spurt and for curves that exceed 30 degrees [5].

Treatment is guided by the severity of deformity and is intended to prevent further curve progression, re-establish trunk balance and symmetry, but also prevent long term complications (pain, loss of mobility, cardiac and pulmonary dysfunctions) [6]. Surgical treatment mainly consists of arthrodesis, and is indicated when the curves exceed 45–50 degrees. There are many types of interventions with multiple approaches, anterior, posterior, or combined anterior and posterior, but also multiple implants. Posterior arthrodesis and segmental instrumentation have been and still are the standard in surgical correction of adolescent idiopathic scoliosis [7].

All vertebral arthrodeses involve both short and long term complications and even though the implants and the surgical techniques have evolved, the complications rate has remained constant [6,8]. Surgeries performed through a posterior approach have a complication rate of 5.1%, 5.2% through an anterior approach, and 10.2% through a combined approach [9]. Neurological complications have a rate of occurance of under 1% while non-neurological complications appear more frequent, ranging between 7% up to 15%. Even though neurological complications are isolated, they are usually severe and consist of nerve root injuries, cauda equina or spinal cord deficit, that can lead to partial or total paralysis, peripheral nerves deficits, or in exceptional cases, death. Among the causes of neurological complications are marrow compression, epidural hematomas or abscesses, or iatrogenic injuries of the nervous elements. In addition, marrow distraction during correction and ischemic lesions that lead to a low blood flow can be causes of neurological complications [10,11,12]. Non-neurological complications, infection, pseudarthrosis, curve progression, and proximal junctional kyphosis, even though more frequent, rarely have a functional impact and a lower risk of re-intervention [10,13].

Rapid correction of rigid curves can increase the risk of neurological complications [14], and traction can be a good orthopaedic technique for progressive deformity correction; scoliosis being one of the spine disorders for which it can be successfully applied, in an attempt to increase correction and minimize complications. An increase in patient satisfaction and quality of life was observed in cases with a high Cobb angle correction [15].

Through traction, a gradual curve correction is achieved in the frontal, sagittal, or axial plane, allowing for a continuous and conscious neurologic monitoring. Traction can be used for all spinal deformities, with few exceptions: cranial malformations that cannot allow for an adequate pin placement, osteogenesis imperfecta considered to be a relative contraindication, and osteoporosis being addressed by placing a higher number of pins that are tightened at a lower torque. Presence of intra or extradural growths and medullar canal stenosis, with or without neurological deficits, are considered absolute contraindications [16].

Halo gravity traction effectiveness is well known but it is not without complications. The most encountered complications are neck pain, headache, pin pain, visual disorders, neurological complications (upper limb numbness, palsy), or pin infections, but there are few studies that address them distinctively [17].

The purpose of this study is to assess the complications that arise during halo gravity traction and to evaluate correction of the scoliotic curves under traction.

## 2. Materials and Methods

We conducted a single centre prospective study on 19 paediatric patients suffering from scoliosis that were admitted in the 2019–2022 period.

The patients included in our study are children suffering from adolescent or juvenile idiopathic scoliosis that had the main curve with a Cobb angle larger than 65 degrees and did not undergo any spine surgeries prior to admission. Exclusion criteria were scoliosis with vertebral malformations, vertebral canal stenosis or growths, cranial malformations, osteoporosis, or osteogenesis imperfecta. 

After admission, all the patients underwent the same series of investigations. Full spine standing X-rays were performed, a head CT in order to determine the thickness of the skull and to allow for the safe placement of pins, and also a spine MRI was taken to evaluate the marrow and marrow canal. After performing the investigations, the halo was applied by a neurosurgeon under general anaesthesia. Patients’ hair was shaved at the pin insertion sites and four pins were used that were tightened to 4 lbs of torque. Three halo gravity traction devices were used for each patient: bed halo gravity-traction, rolling chair, and walker halo gravity traction, the three devices allowing the patient to have good mobility (Figure 1). Traction was started 2 days after the halo was applied by adding 1 kg (kilogram) weights on each device. It was permanently maintained, except during meals and personal hygiene. The weight was increased daily until at least 40% to a maximum of 50% of the patient’s weight was achieved. Pin tension was verified at 48 h, 72 h post halo application and weekly during traction. Patients were educated for daily pin care, using sterile cotton swabs with saline solution around the pins and alcohol or chlorhexidine-alcohol solutions for the halo, also they were instructed in reporting any symptoms related to traction: pin secretions, pin pain, cervical pain, back pain, headache, vertigo, or neurological signs. Psychological support was offered for all the patients since day one, they were counselled in how to socially cope with the halo traction device and face, what might have been, uncomfortable questions or staring. Patients were also instructed for proper hair and scalp hygiene. All nursing staff and medical officers were educated on the traction devices and evaluation of pin site infections or neurological changes (upper limb numbness or palsy).

Patient details included patient age, gender, weight, traction weight, and curve type. Traction related complications were recorded (cervical pain, headache, pin pain, vertigo, back pain, neurological disorders, pins infection, and pins displacement). Radiological evaluation was performed by measuring the main curves using standing front view X-rays, the maximal coronal Cobb angles were evaluated before traction and preoperative (post traction), and each curve was classified using Lenke classification (Figure 2). The same examiner performed all measurements. 

The statistical data were processed using SPSS (IBM Corp., Armonk, NY, USA) and Microsoft Office Excel. Statistical Analysis Descriptive statistics were calculated in terms of means and standard deviations for continuous variables; frequencies and percentages were calculated for discrete variables. 

## 3. Results

Patients had a mean age of 14.16 years, 68.4% girls and 31.6% boys. The mean traction duration was 28.46 days. Patients’ mean body weight was 48.89 kg and the mean traction weight was 20.11 kg, representing 41.13% of body weight (Table 1). Patients that underwent traction for under 25 days had a mean angle correction of 32.32% and the ones that were under traction for more than 25 days had a mean angle correction of 49.96% (Figure 3). 

Regarding the main scoliotic curve, the most frequent localisation was thoracic 52.63%, only one patient had a lumbar curve, while 42.11% of the patients had thoraco-lumbar curves. The mean Cobb angle before traction was 86.21 degrees and 63.16 degrees after traction, thus achieving a correction of 26.73%. Skeletal maturity was determined using the Risser sign; 42.1% of the patients were classified as Risser 4 and 15.8% Risser 0. Vertebral rotation was assessed and 47.4% of the patients were classified as grade III according to the Nash and Moe system; 42.1% had grade II rotation and 10.5% (Table 2).

The majority of the curves, 42.1%, were Lenke type 3, 36.8% type 1, while only 5.3% of the curves were classified as type 2 and type 6. Regarding the thoracic sagittal profile, 73% had a normal sagittal profile. The majority of the patients (47.4%) had a type A lumbar spine modifier (Table 3). The highest curve correction was achieved in type 3 curves (29%), followed by type 1 (28%). The lowest correction was for the patient with a type 2 curve (10%), patients with type 5 curves experiencing a correction of 21%, and patients with type 6 had an improvement of 19% (Figure 4). In relation to skeletal maturity, the greatest correction was achieved for patients with Risser 0 (32%), followed by the patients with Risser 5 (31%). The lowest correction was observed in patients with Risser 2, while patients with Risser 3 had an improvement of 27%, and patients with Risser 4 experienced a correction of 26% (Figure 5). 

Complications were encountered in 94.7% of the patients, the most frequent being cervical pain that was present in 89.5% of the cases. It was followed by back pain in 36.8% of the cases, just 5.3% of the cases having experienced vertigo or pin displacement. Neurological symptoms were present in 26.3% of the patients. Pin pain and pin infection affected 26.3% of the patients. Headache was observed in 10.5% of the patients (Table 4). Patients who did not experience any cervical or back pain showed the most improvement, with a mean angle correction of 27% for both patient groups (Figure 6 and Figure 7). Conversely, patients who experienced pin pain and headache had the most curve correction, 31% for the patients with pin pain and 32% for the patients with headache (Figure 8 and Figure 9). 

## 4. Discussion

Scoliotic deformities can have a negative impact on respiratory compliance, thoracic cage, and skeletal and muscular function [18]. 

Nickel and Perry have used, for the first time, halo traction in the 1960′s at the “Rancho Los Amigos” hospital and revolutionized the management of spinal deformities [19]. In the 1970′s, Stagnara used the halo to develop gravity traction and used it to correct spine deformities [20]. The halo is also used in other forms of skeletal traction: halo-femoral, halo-tibial, and halo-pelvic [19]. Halo gravity traction is a useful preoperative technique that can gradually reduce deformity and provide a safe means of correction, reducing neurological risk [21].

Traction duration is yet to be agreed upon, with reports ranging from weeks to months [22,23,24,25]. Letts et al. analysed the time dependent correction of halo femoral traction in eight patients, achieving a maximum correction (40%) in the first week, recommending traction to be maintained for three weeks [24,25,26]. Park et al., in their study on 20 paediatric patients with scoliosis, concluded that the maximum correction was obtained in the first two weeks and complications may increase with prolonged traction [27]. Bogunovic et al. obtained a maximum correction in an average of 42.6 days and the mean curvature (96.48) of the 33 patients was corrected to 66% after 2 weeks and to 91% after 3 weeks [28]. Rocos et al. applied a protocol with weight addition to 50% of body weight at 3 weeks and traction was kept in place until signs of neurological complications occurred or a maximum of 6 weeks had passed. The study was performed on 24 patients with a mean age of 11.8 years. The most improvement (82%) occurred in the first 3 weeks and the mean duration of traction was 42 days [29]. Hwang et al. obtained the most correction (28.2%) within 1 week and observed that significant changes in the curvature, a correction of 34%, occurred in up to 2 weeks [30].

Patients included in our study had a mean traction duration of 28.46 days, ranging from 17 to 43, with a mean and maximum improvement of 26.73% and 46.95%. The most correction was obtained in patients that underwent traction for at least 25 days, patients with type 3 Lenke curves achieving the highest correction of 29%. 

Halo-gravity traction, even though considered a safe and effective method, is not without risks. Pin loosening, pin infections, and gastrointestinal discomfort have been reported [30,31]. In their study, Iyer et al. reported complications in 9 of their 30 patients, all of them infection-related an no neurological complications [32]. Bogunovic et al. had a complication rate of 27% (transient nystagmus, upper extremity numbness, pin site erythema/infection, unilateral miotic pupil, and progression of myelopathy) and they used a traction weight for each patient of 33.5% of their body weight [28]. Rocos et al., of the 24 patients, had 1 patient with early signs of cranial nerve palsy, 8 patients with pin loosening, and one with transitory urinary disturbance [29]. In their study of 59 patients, Hwang et al. recorded complications in 10 cases: vomiting, intolerable pin site pain, pin site infection, skull fracture, and pelvic wound [30].

In contrast to other reported series, we encountered complications in most of our patients (94.7%), but they were only temporary and easily addressed. The most frequent was cervical pain, which mainly occurred due to neck muscle stretching and it was resolved by maintaining or decreasing traction weight for 1 or 2 days. Patients received oral anti-inflammatory drugs and anti-inflammatory gels locally. Back pain was resolved by using the same treatment and it was present in seven cases. 

Neurological symptoms (upper limb numbness or palsy), pin pain, and pin infection were found in 26.3% of the patients. Even though each patient was instructed for daily halo care, superficial pin infections still appeared, but they were quickly resolved by locally applying Iodine solutions; no cerebral abscess or osteitis were identified. Pin pain can be a consequence of pin loosening, and every time it was reported, pin tension was checked. It was not persistent and most of the times diminished a few hours after pin tightening and oral anti-inflammatory drugs (Ibuprofen). Patients included in our study had only mild neurological symptoms (upper limb numbness or transient palsy), and each time they occurred a neurological exam was performed. They were addressed by decreasing traction by at least 2 kg and using oral anti-inflammatory medication. Headache was present in 10.5% of the patients and usually occurred the next day after halo installation, before applying traction. It alleviated progressively as the patients familiarised with their new condition. One of the patients experienced vertigo, but the neurologic exam did not reveal any deficits. We also had one patient who experienced pin displacement, even though we followed the same protocol for tightening and monitoring the pins. This can be due to the fact that the patient could not tolerate sleeping on his back and preferred his sides, thus applying uneven pressure on the pins. 

One of the biggest drawbacks of the study is the small sample size, which did not allow us to find a direct correlation between complications and traction duration, however, we still find the study informative as we could not find similar series that specifically address traction-related complications. 

## 5. Conclusions

The main goal of halo gravity traction is to increase patient safety by reducing intraoperative risk, but it does not come without complications. The most feared are neurological and deep infectious complications, mostly due to their slow regression and their potential of rapid decline. Despite our data suggesting that most of the patients experienced complications (94.7%), they were not severe, and manageable with minimum intervention, with the most frequent being cervical pain. Infectious complications were only superficial, with no long-term consequences, as were neurological complications. Although complications such as cervical pain, back pain, neurological symptoms, pin pain, and pin infection are frequent, with proper patient monitoring they can be addressed safely with minimum consequences. The use of various mobility devices reduces patient discomfort and makes the method a safe means of obtaining up to 50% of curve correction in certain cases. We found that applying the same traction protocol allowed us to better monitor the patients and provide them better care. A future larger and more homogenous study would be necessary in order to better understand the relationships between complications and traction duration, curve correction and patient age. 

## Figures and Tables

**Figure 1 children-09-01701-f001:**
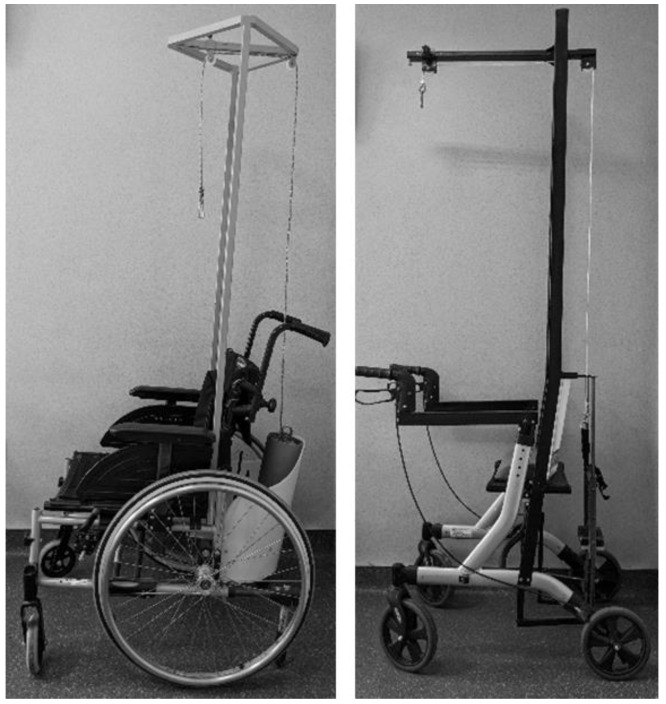
Wheelchair and walker.

**Figure 2 children-09-01701-f002:**
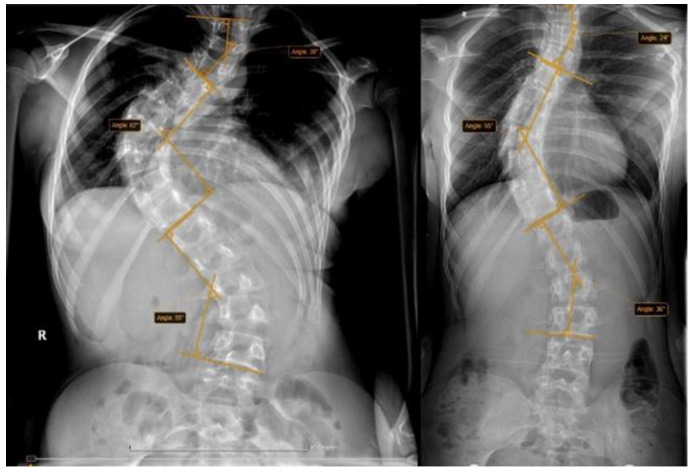
Pre-traction and pre-operative standing X-rays of the same patient.

**Figure 3 children-09-01701-f003:**
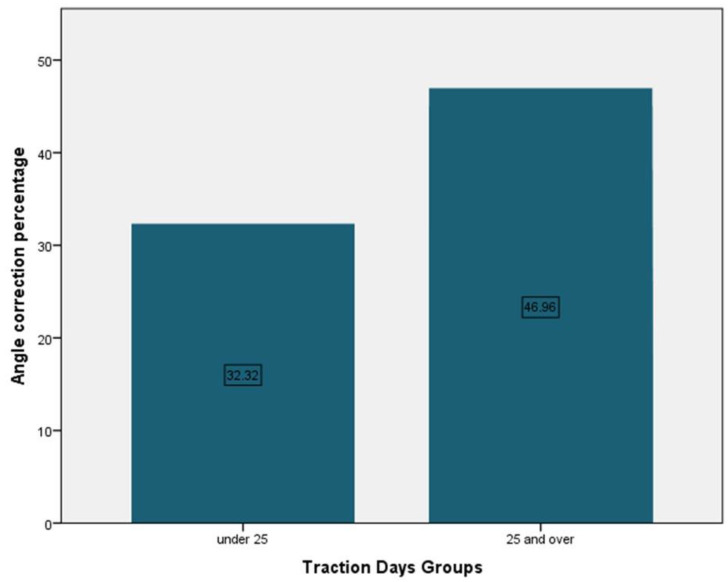
Traction days mean angle correction.

**Figure 4 children-09-01701-f004:**
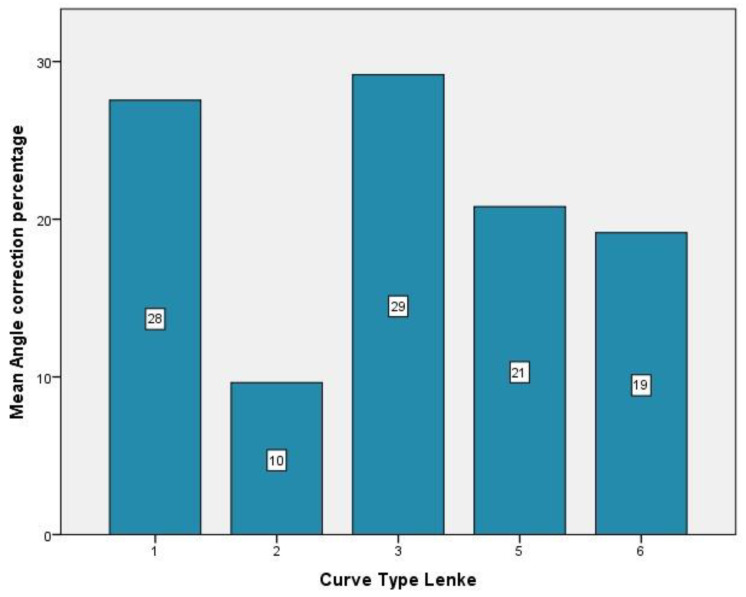
Lenke type mean angle correction.

**Figure 5 children-09-01701-f005:**
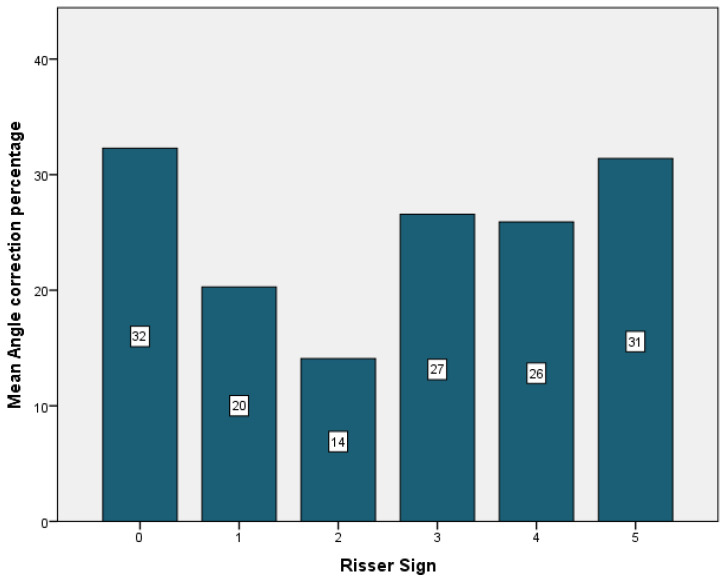
Risser sign mean angle correction.

**Figure 6 children-09-01701-f006:**
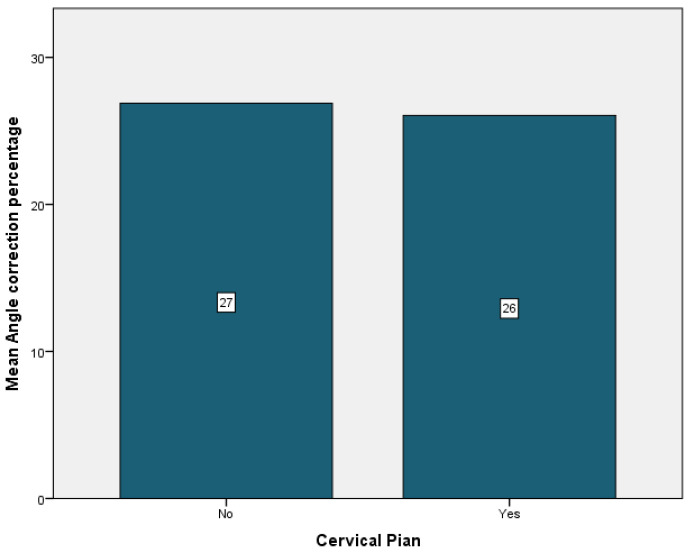
Cervical pain mean angle correction.

**Figure 7 children-09-01701-f007:**
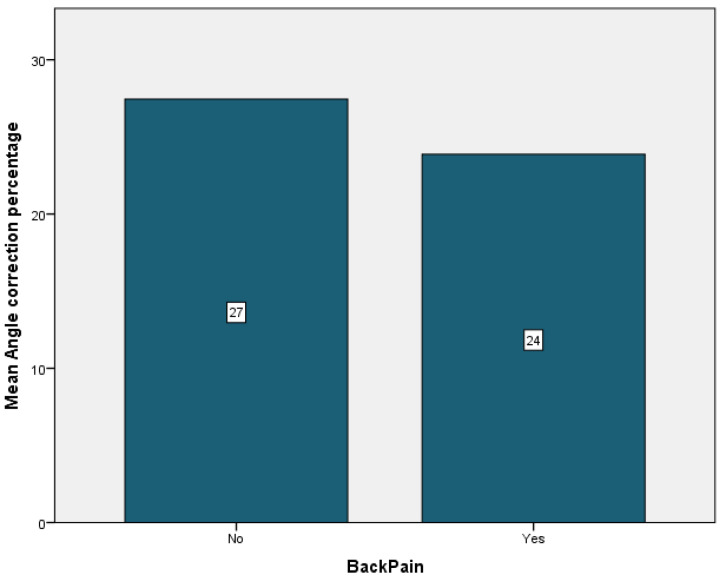
Back pain mean angle correction.

**Figure 8 children-09-01701-f008:**
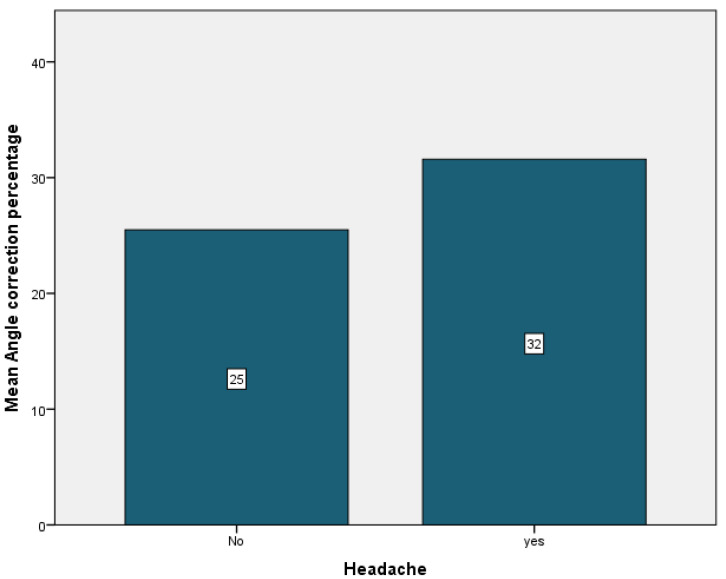
Headache mean angle correction.

**Figure 9 children-09-01701-f009:**
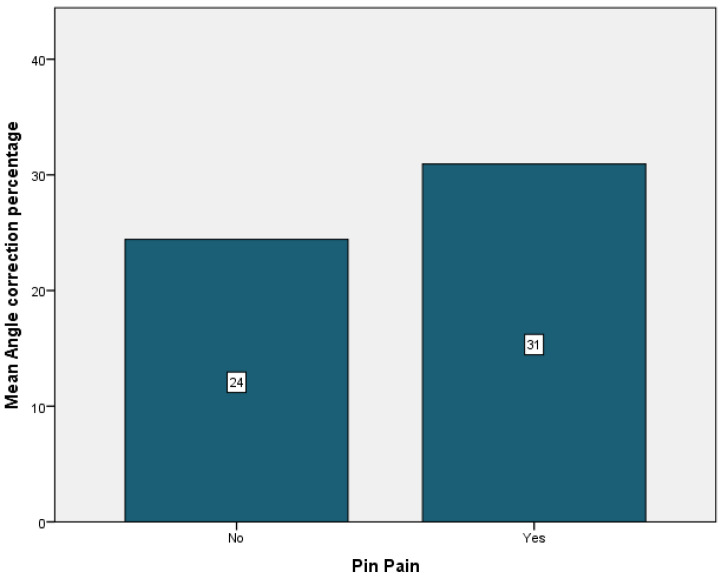
Pin pain mean angle correction.

**Table 1 children-09-01701-t001:** Demographic data.

Variable	Study Group (n = 19)
Age (years)	14.16 * ± 2.46 ** (range 9–17)
Gender n, (%)	
Males	6 (31.6)
Females	13 (68.4)
Body Weight	48.89 * ±9.83 ** (range 25–63)
Traction duration (days)	28.46 * ±7.86 ** (range 17–43)
Traction weight (Kg)	20.11 * ±2.54 ** (range 12–22)

* Mean, ** ± Standard deviation.

**Table 2 children-09-01701-t002:** Curve information.

Variable	Study Group
Main curve localisation n, (%)	
Thoracic	10 (52.63)
Lumbar	1 (5.26)
Thoraco-lumbar	8 (42.11)
Cobb angle before traction (degrees) Mean ± Standard deviation	86.21 ±13.88 (range 66–115)
Cobb angle pre-operative (degrees) Mean ± Standard deviation	63.16 ± 10.27 (range 47–85)
Vertebral rotation n, (%)	
++	8 (42.1)
+++	9 (47.4)
++++	2 (10.5)
Risser sign n, (%)	
0	3 (15.8)
1	2 (10.5)
2	1 (5.3)
3	4 (21.2)
4	8 (42.1)
5	1 (5.3)

**Table 3 children-09-01701-t003:** Lenke classification.

Variable n, (%)	Study Group
Curve type	
1	7 (36.8)
2	1 (5.3)
3	8 (42.1)
5	2 (10.5)
6	1 (5.3)
Lumbar spine modifier	
A	9 (47.4)
B	7 (36.8)
C	3 (15.8)
Thoracic sagittal Profile	
Normal	14 (73.7)
+	2 (10.5)
-	3 (15.8)

**Table 4 children-09-01701-t004:** Complications rate.

Variable n, (%)	Study Group (n = 19)
Complication occurrence	18 (94.7)
Cervical pain	17 (89.5)
Back pain	7 (36.8)
Neurological symptoms	5 (26.3)
Headache	2 (10.5)
Vertigo	1 (5.3)
Pin pain	5 (26.3)
Pin infection	5 (26.3)
Pin displacement	1 (5.3)

## Data Availability

Data available on request.

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
