# Peer review of "The Use and Complications of Halo Gravity Traction in Children with Scoliosis"

_children, 2022, doi:10.3390/children9111701_

Round 1

Reviewer 1 Report

Dear Sir/Mam

Please find bellow the requested review regarding the manuscript. The article contains a lot of useful information on the issue. The topic is very interesting and but use of sources is not appropriate. Although it has some useful information there are less references and the statements are not established. I suggest the authors to write more information with references.

The abstract is too brief and introduction section involves too many information. The research question is not justified clearly, given what is already known about the topic. The results are not discussed from multiple angles and conclusions answer the aims of the study partially. The conclusions are partially supported by references or results.

Author Response

Thank you for your comments.

              We have gone through your comments carefully and tried our best to address as many of them as we could. We managed to write more information with references. As far as the abstract goes, we were obliged to meet the journal’s requirements . The introduction section was addressed and alterations were made. The discussions and conclusions were revised and we hope to meet your expectations. We thank you for your careful and thorough reading and for the constructive suggestions which helped us improve the manuscript and better define our work. We hope the manuscript has been improved accordingly.

Reviewer 2 Report

Complications that arise during halo gravity traction in idiopathic scoliosis patients.  In this manuscript, the authors studied the complications and evaluated the correction of the scoliotic curves under traction in 19 pediatric patients. Traction-related complications were encountered in 94.7% of the patients, the most frequent being cervical pain which was present in 89.5% of the cases. It was followed by back pain, in 36.8% of the cases, 5.3% of the cases having experienced vertigo or pin displacement. Neurological symptoms were present in 26.3% of the patients. Pin pain and pin infection also affected 26.3% of the patients.  This manuscript is compelling and interesting, the data is clear, and the discussions are adequate.

To make the manuscript more profound, I would like to give three suggestions:

1.     Please provide a background description of the genetic and environmental risk factors for idiopathic scoliosis.

2.     Could authors provide the control data in patients without halo gravity traction? Comparison of complications and assessment of scoliosis correction in patients treated with halo-gravity traction and untreated patients.

3.     Please change the comma to the point. Such as line 25 is 0.47% instead of 0,47%. Table 1 has the same problem.

Author Response

Thank you for taking the time to review our paper, for your comments regarding this article and we appreciate the positive feedback.

  1. We have modified the introduction and added information regarding risk and environmental factors for idiopathic scoliosis.
  2. Regarding the comparison control group, we could not establish one as in the last years in our clinic, most of the patients with severe idiopathic scoliosis have been operated after being subjected to halo gravity traction and the remaining patients did not meet the inclusion criteria.
  3. Thank you for pointing the errors. We did our best to address them.

We hope that these revisions are sufficient to make our manuscript suitable for publication

Round 2

Reviewer 1 Report

I agree the article to  be accepted in current form.